# Early outcome prediction with quantitative pupillary response parameters after out-of-hospital cardiac arrest: A multicenter prospective observational study

**Tomoyoshi Tamura**[1], **Jun Namiki**[1,2]*, **Yoko Sugawara**[3], **Kazuhiko Sekine**[3], **Kikuo Yo**[4], **Takahiro Kanaya**[5], **Shoji Yokobori**[5], **Takayuki Abe**[6], **Hiroyuki Yokota**[5], **Junichi Sasaki**[1]

1 Department of Emergency and Critical Care Medicine, Keio University School of Medicine, Tokyo, Japan, 2 Department of Emergency Medicine, KKR Tachikawa Hospital, Tokyo, Japan, 3 Department of Emergency and Critical Care Medicine, Tokyo Saiseikai Central Hospital, Tokyo, Japan, 4 Department of Emergency and Critical Care Medicine, Hiratsuka City Hospital, Kanagawa, Japan, 5 Department of Emergency and Critical Care Medicine, Nippon Medical School, Tokyo, Japan, 6 Clinical and Translational Research Center, Keio University, Tokyo, Japan

* namiki@med.keio.ac.jp

**Data Availability Statement:** All relevant data are within the manuscript and its Supporting Information files.

## Abstract

We aimed to determine the characteristics of quantitative pupillary response parameters other than amplitude of pupillary light reflex (PLR) early after return of spontaneous circulation (ROSC) and their implications for predicting neurological outcomes early after cardiac arrest (CA). Fifty adults resuscitated after non-traumatic out-of-hospital CA from four emergency hospitals were enrolled. Pupil diameters, PLR, constriction velocity (CV), maximum CV (MCV), dilation velocity (DV), latency of constriction, and Neurological Pupil index (NPi) were quantitatively measured at 0, 6, 12, 24, 48, and 72 h post-ROSC using an automated pupillometer. Change over time of each parameter was compared between favorable (Cerebral Performance Category [CPC] 1 or 2) and unfavorable neurological outcome (CPC 3–5) groups. Prognostic values of 90-day favorable outcome by these parameters and when combined with clinical predictors (witness status, bystander cardiopulmonary resuscitation, initial shockable rhythm, implementation of target temperature management) were tested. Thirteen patients achieved favorable outcome. CV, MCV, DV (P < 0.001), and NPi (P = 0.005) were consistently greater in the favorable group than in the unfavorable outcome group. Change over time was not statistically different between the groups in all parameters. CV, MCV, DV (ρ = 0.96 to 0.97, P < 0.001, respectively), and NPi (ρ = 0.65, P < 0.001) positively correlated with PLR. The prognostic value of 0-hour CV (area under the curve, AUC [95% confidence interval]: 0.92 [0.80–1.00]), DV (0.84 [0.68–0.99]), and NPi (0.88 [0.74–1.00]) was equivalent to that of PLR (0.84 [0.69–0.98]). Prognostic values improved to AUC of 0.95–0.96 when 0-hour PLR, CV, DV, or NPi was combined with clinical predictors. The 0-hour CV, MCV, and NPi showed equivalent prognostic values to PLR alone/in combination with clinical predictors. Using PLR among several quantitative pupillary response parameters for early neurological prognostication of post-CA patients is a simple and effective strategy.

**Funding:** JN received grant from Japan Society for the Promotion of Science (KAKENHI 15H05009; https://www.jsps.go.jp/english/). The funders had no role in study design, data collection and analysis, decision to publish, or preparation of the manuscript.

**Competing interests:** The authors have declared that no competing interests exist.

## Introduction

The incidence of neurological sequelae among survivors of out-of-hospital cardiac arrest (OHCA) remains as high as 75% despite advances in post-arrest care [1]. Moreover, early mortality after OHCA is reported to be the result of withdrawal of life-sustaining therapy (WLST) because of perceived unfavorable neurological prognosis [2, 3]. In comatose post-cardiac arrest (CA) patients, bilateral absence of pupillary reflex to light at 72 h or longer after CA strongly predicts poor neurological outcome [4]. In the last several years, quantitative assessment of pupillary light reflex (PLR) has emerged as a promising modality for predicting neurological outcomes in post-CA patients. Several publications reported the prognostic superiority of the quantitative assessment of PLR over the qualitative evaluation of PLR before 72 h after CA [5–9]. Recently, we have reported that quantitative values of PLR in survivors or in patients with favorable neurological outcomes were consistently greater than values in non-survivors or in patients with unfavorable outcomes, respectively, within 72 h following the return of spontaneous circulation (ROSC) after OHCA [9]. Notably, quantitative PLR strongly predicted 90-day survival and favorable neurological outcomes as early as 0 h after ROSC.

Pupillary response is affected by various factors including age, drugs commonly used in post-arrest care, or therapeutic hypothermia [10, 11]. Although qualitative pupillary examination is routinely performed in post-CA patients, little is known about the change in pupillary response parameters early after ROSC. Among other quantitative pupillary parameters, the prognostic accuracy of quantitative PLR is mainly studied in post-CA patients [5–9, 12, 13]. Consequently, the prognostic implication of other parameters that could be simultaneously measured by a quantitative pupillometer is unclear and has never been compared.

The primary aim of this study was to clarify a role of quantitative pupillary responses that can be measured by a pupillometer within the first 72 h after ROSC for early neurological prognostication of post-CA patients by investigating the optimal parameter among all quantitative pupillary response parameters as well as their consistency during 72 h after ROSC. The secondary aim was to test the hypothesis that a combination of quantitative pupillometry and clinical information, which is used as a predictor of outcomes in previous studies, achieves the best prognostic performance for 90-day favorable neurological outcome.

## Methods

### Study design and setting

This study is a post-hoc analysis of a multicenter, single-arm, uncontrolled, prospective, observational study that focused on quantitative PLR as a representative parameter of pupillary responses (clinical trial identifier: UMIN000015658) [9]. This study was approved by the IRB of each participating institution (Keio University School of Medicine, Ethics Committee; Institutional Review Board of Nippon Medical School; Research Ethics Committee, Tokyo Saiseikai Central Hospital; and Independent Ethics Committee of Hiratsuka City Hospital). Written informed consent was obtained from the patients' family. The study methodology is compliant with STARD guidelines [14].

### Selection of patients and data collection

Fifty adult OHCA patients (≥18 years old) who achieved ROSC and did not meet the exclusion criteria–CA due to obvious trauma, a do-not-attempt-resuscitation order, pregnancy, dependence on others for daily support because of impaired brain function, terminal stage of a known malignancy which makes 3-month survival unlikely, and use of extracorporeal membrane oxygenation–were prospectively enrolled in four university hospitals and emergency

medical centers. Data collected were age, sex, witness status, presence of bystander cardiopulmonary resuscitation (CPR), initial arrest cardiac rhythm, location of CA, etiology of CA, time from collapse to ROSC, Glasgow Coma Scale (GCS) after ROSC in the emergency department (ED), and implementation of target temperature management (TTM).

Comatose (GCS ≤8 with motor response of ≤5) patients received standard post-arrest care according to international guidelines [4, 15]. Briefly, patients presenting GCS ≤8 with motor response of ≤5 were intubated and mechanically ventilated. Midazolam, propofol, dexmedetomidine, and fentanyl or buprenorphine were used for sedation and analgesia. Neuromuscular blocking agents, such as rocuronium or vecuronium, were used in adjunct to treat shivering. Norepinephrine, dopamine, and/or dobutamine were given to maintain optimum tissue perfusions at the discretion of the treating physicians. Patient body temperature was managed at 33–36˚C (TTM) or with fever control for non-cardiogenic CA due to subarachnoid hemorrhage or sepsis. Non-comatose (GCS >8 in the ED) post-CA patients were also enrolled but did not undergo therapeutic hypothermia [9].

## Quantitative measurements of pupillary responses

Pupillary examinations were performed using an automated quantitative pupillometer (NeurOptics® NPi™-100 pupillometer, Neuroptics Inc., Irvine, CA, USA) [5]. Parameters were as follows: maximum diameter (MAX; mm), baseline pupil size before constriction; minimum diameter (MIN; mm), pupil diameter at peak constriction; latency of constriction (LAT; msec), time of onset of constriction following initiation of the light stimulus; constriction velocity (CV; mm/sec), average velocity of pupil constriction of the diameter; maximum constriction velocity (MCV; mm/sec), maximum velocity of pupil constriction of the diameter; dilation velocity (DV; mm/sec), average velocity of pupil dilation back to the initial resting size; and NPi, a proprietary algorithm that takes all variables as inputs and compares them to a normative model to give a composite score between 0 and 5 [16]; as well as the amplitude of the PLR, percent change between MAX and MIN. Quantitative measurements were performed at 0, 6, 12, 24, 48, and 72 h after ROSC. The largest value of these measurements for both eyes at each time point was adopted for the analysis. A value of 0 was imputed for CV, MCV, DV, and NPi for patients with 0% PLR.

## Outcomes

The outcome variable was the 90-day neurological outcome assessed by the Cerebral Performance Category (CPC) scale. The CPC scale categorizes neurological outcomes as follows: CPC 1, good performance; CPC 2, moderate disability; CPC 3, severe disability; CPC 4, comatose or persistent vegetative status; and CPC 5, brain death or death. Neurological outcome was dichotomized as favorable (CPC of 1–2) or unfavorable (CPC of 3–5) neurological outcome [17].

## Analysis

Descriptive statistics, comparisons of binary variables, mixed-effects model for the differences of repeated measurements between the outcome groups, and receiver operating characteristics (ROC) curve analysis for the comparison of the area under the curve (AUC) values were performed as previously reported [9]. Mann-Whitney *U*-test was used for the comparison of differences between not normally distributed groups. Pearson correlation coefficient was calculated to determine the correlation between parameters measured with a pupillometer. A multivariable logistic regression model was used with adjustment for known potential covariates to analyze the associations. A set of potential confounders was chosen as clinical predictors

based on published literature including witness status, bystander CPR, initial shockable rhythm, and implementation of TTM [4, 18–20]. Prognostic values of models, clinical predictors alone, and combination of clinical predictors and 0-hour PLR, CV, DV, or NPi were compared. Moreover, a multivariable logistic regression with backward elimination method was used with adjustment for known potential covariates to analyze the associations between 0-hour PLR, CV, DV, or NPi and the primary outcome. Independent variables selected for adjustment were witness status, bystander CPR, initial shockable rhythm, implementation of TTM, and 0-hour PLR. P values <0.05 were considered statistically significant. All analyses were conducted using IBM SPSS version 25.0 (IBM Corp., Armonk, NY, USA).

## Results

### Characteristics of study subjects

Baseline patient characteristics are shown in Table 1. Three non-comatose patients (2 patient with GCS 10, and 1 patient with GCS 15) were included in this study. Compared to patients with unfavorable neurological outcome, 13 (26%) patients who achieved favorable neurological outcome, were significantly younger (P = 0.001) and had more episodes of CA in a public location (P = 0.003), more initial shockable rhythms (P < 0.001), high presumed cardiac cause (P = 0.04), shorter CA duration (P < 0.001), higher rate of achieving prehospital ROSC (P = 0.002), and higher rate of TTM implementation (P = 0.04).

### Quantitative measurements of pupillary responses during 72 h after ROSC

CV (P < 0.001), MCV (P < 0.001), DV (P < 0.001), and NPi (P = 0.005) as well as PLR (P < 0.001) [9] were consistently greater among patients with favorable neurological outcomes than those with unfavorable outcomes at each time point during the first 72 h. On the contrary, no difference in MAX, MIN, and LAT was observed between the two outcome groups. Similar

**Table 1. Patient characteristics.**

| | Favorable neurological outcome (N = 13) | Unfavorable neurological outcome (N = 37) | P value |
|---|---|---|---|
| Age; median y/o (IQR) | 54 (37–62) | 71 (56–81) | 0.001 |
| Male sex; n (%) | 12 (92) | 24 (65) | 0.08 |
| Witness status; n (%) | 9 (75) | 27 (73) | 1.00 |
| Bystander performed CPR; n (%) | 7 (54) | 17 (47) | 0.75 |
| CA in public location; n (%) | 12 (92) | 15 (42) | 0.003 |
| Initially in shockable rhythm; n (%) | 9 (69) | 5 (14) | < 0.001 |
| Presumed cardiac cause; n (%) | 11 (85) | 18 (50) | 0.04 |
| Cardiac arrest time; median min (IQR) | 15 (9–18) | 41 (31–52) | < 0.001 |
| Prehospital ROSC; n (%) | 10 (77) | 9 (24) | 0.002 |
| GCS after ROSC in ED; median (range) | 3 (3–15) | 3 (3–5) | 0.06 |
| Implementation of TTM 33°C to 36°C; n (%) | 12 (92) | 22 (61) | 0.04 |
| 90-day CPC | | | |
| 1 | 13 (26) | | |
| 2 | 1 (2) | | |
| 3 | | 1 (2) | |
| 4 | | 8 (16) | |
| 5 | | 27 (54) | |

CA, cardiac arrest; CPC, cerebral performance category; CPR, cardiopulmonary resuscitation; ED, emergency department; EMS, emergency medical service; GCS, Glasgow Coma Scale; IQR, interquartile range; PEA, pulseless electrical activity; TTM, target temperature management; VF, ventricular fibrillation; VT, ventricular tachycardia.

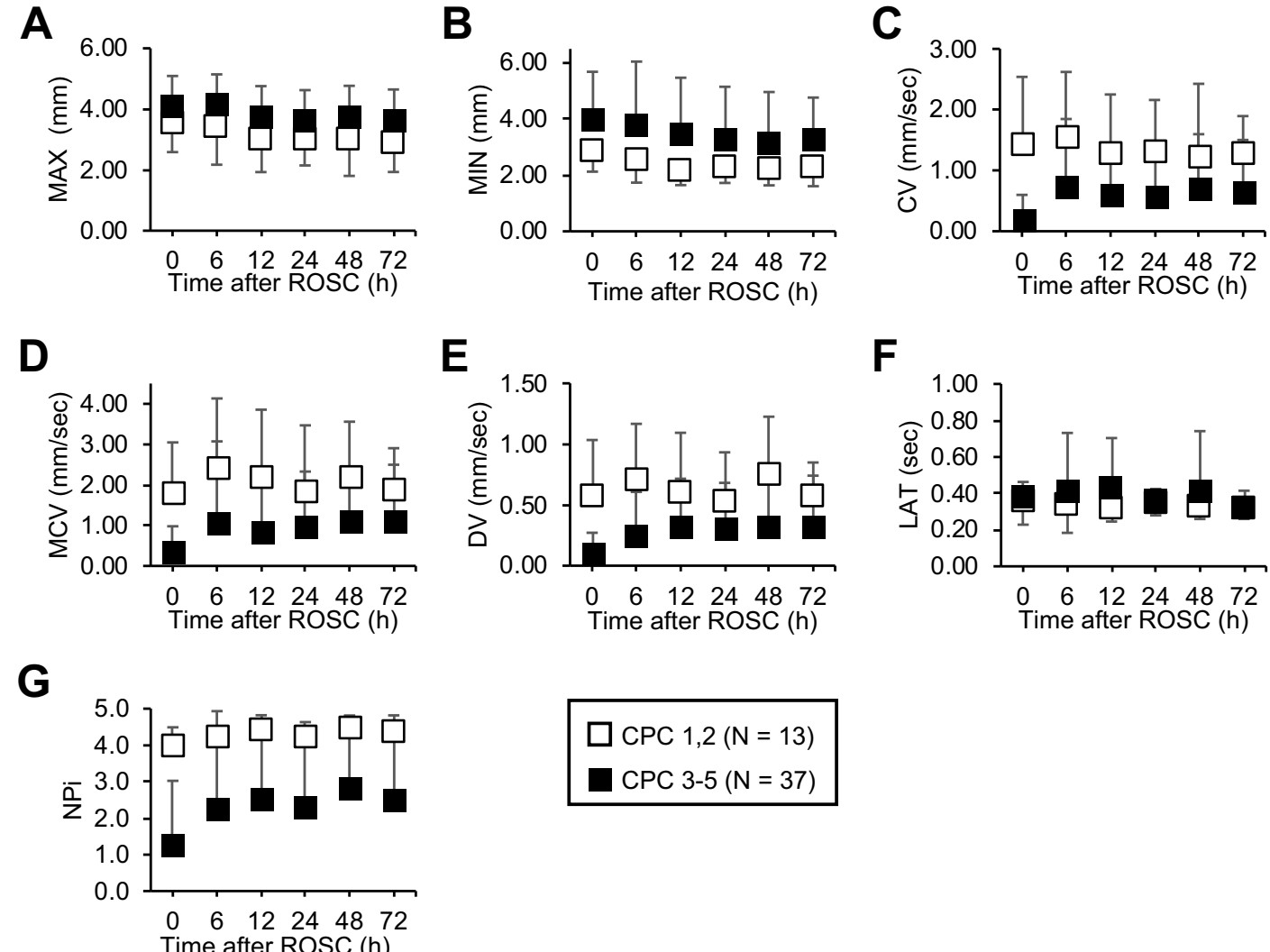

**Fig 1. Serially measured quantitative pupillary responses of favorable and unfavorable neurological outcomes after return of spontaneous circulation to 72 h.** (A) MAX, (B) MIN, and (F) LAT was not different among favorable and poor neurological outcome groups. In contrast, (C) CV, (D) MCV, (E) DV, and (G) NPi were significantly greater in the favorable neurological outcome group compared to poor outcome group. Moreover, the change in each parameter over time was not different between the two outcome groups. Data are shown as mean ± SD. CV, constriction velocity; DV, dilation velocity; LAT, latency of constriction; MAX, maximum diameter; MCV, maximum constriction velocity; MIN, minimum diameter; NPi, Neurological Pupil index; PLR, amplitude of pupillary light reflex.

to PLR (P = 0.86) [9], after accounting for the interaction at a certain time point by outcome status, none of the pupillary parameters were significant (MAX, P = 0.17; MIN, P = 0.86; LAT, P = 0.82; CV, P = 0.39; MCV, P = 0.63; DV, P = 0.79; and NPi, P = 0.91). This indicates that there is no difference in the change of each pupillary parameters over time regardless of outcomes (Fig 1, S1 Table).

Correlations between PLR and other pupillary parameters were strong in CV, MCV, and DV (correlation coefficient [ρ] = 0.96–0.97) and moderate in NPi (ρ = 0.65) (Fig 2). However, the correlation between PLR and MAX, MIN, or LAT was weak (|ρ| ≤ 0.32). This indicated that a positive correlation with PLR was found in pupillary response parameters that were consistently greater in the favorable neurological outcome group than in the unfavorable neurological outcome group.

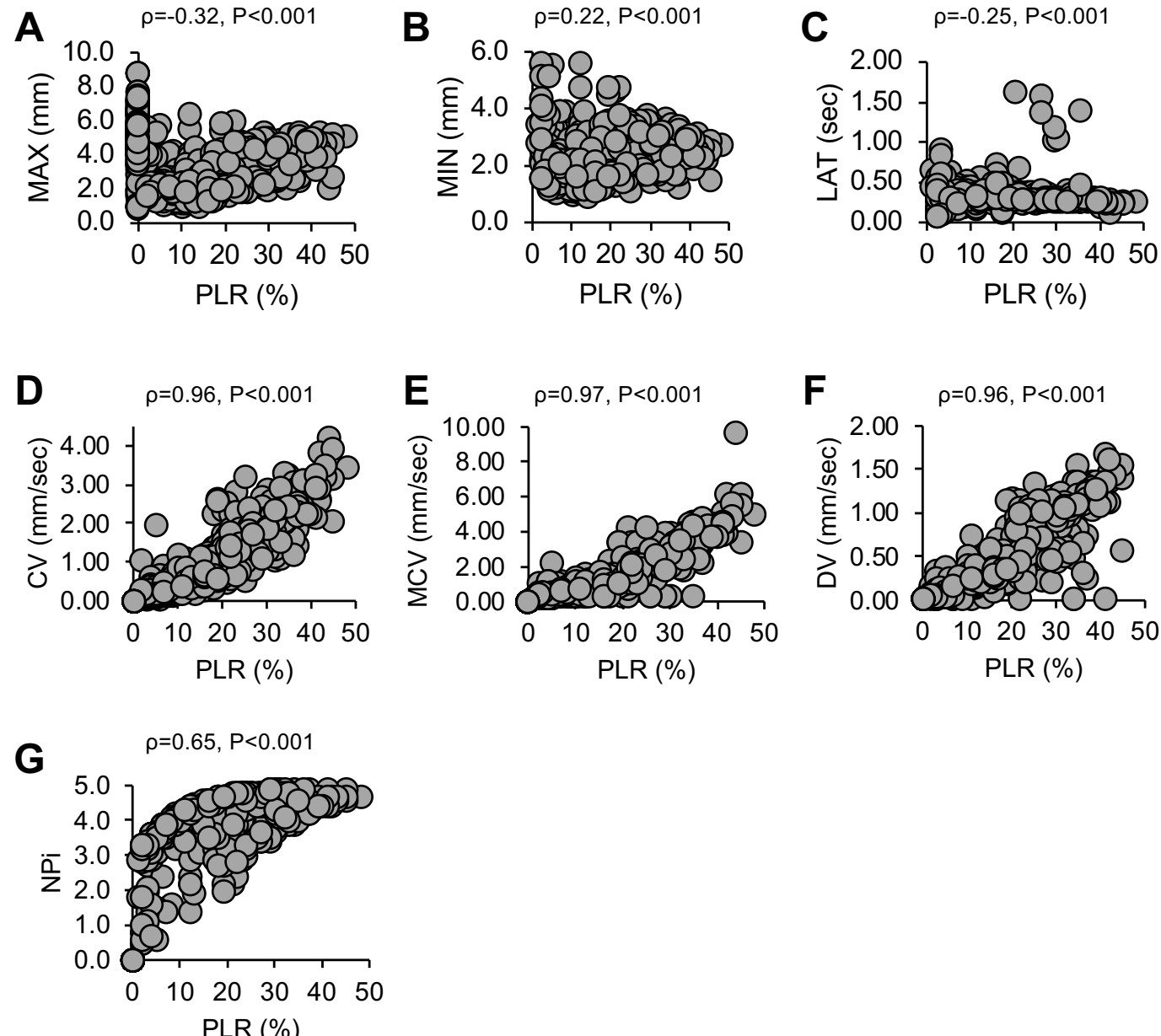

**Fig 2. Correlation between PLR and other quantitatively measured parameters.** Parameters which two outcome groups did not show difference: (A) MAX, (B) MIN, and (C) LAT, showed low correlation with PLR. In contrast, parameters that were greater among favorable outcome groups were (D) CV, (E) MCV, (F) DV, and (G) NPi and showed a high positive correlation with PLR, CV, constriction velocity; DV, dilation velocity; LAT, latency of constriction; MAX, maximum diameter; MCV, maximum constriction velocity; MIN, minimum diameter; NPi, Neurological Pupil index; PLR, amplitude of pupillary light reflex.

### Prediction of 90-day favorable neurological outcome by quantitative pupillary response

The ROC analysis revealed that CV, MCV, DV, and NPi, which were associated with favorable neurological outcome and showed positive correlation with PLR, best predicted favorable neurological outcome with 0-hour value among measurements throughout 72 h: the AUC values of 0-hour CV (0.92), DV (0.84), and NPi (0.88) were comparable to that of 0-hour PLR (0.84)

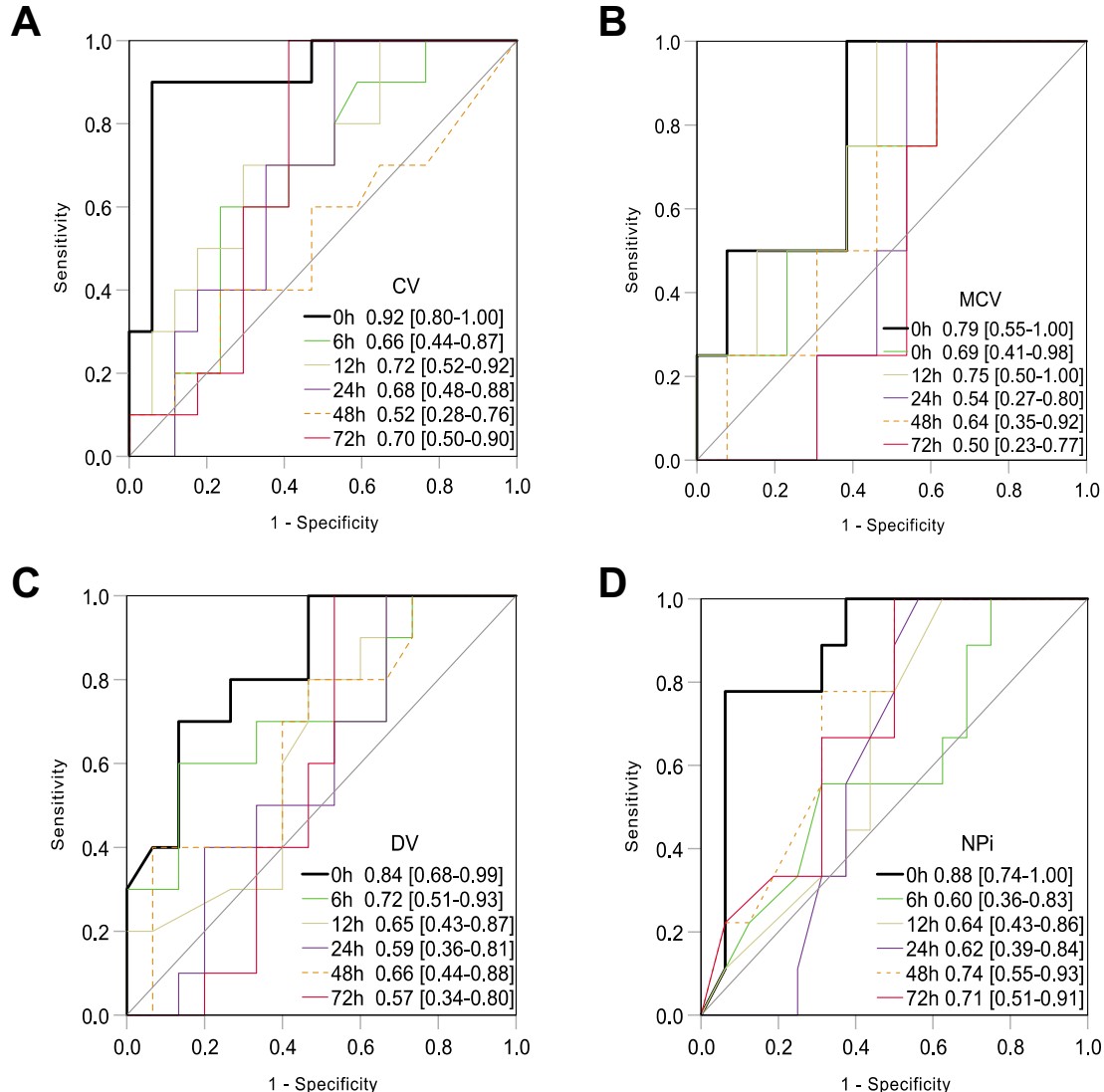

**Fig 3. ROC analysis for prediction of 90-day favorable neurological outcome.** Values represent the area under the curve [95% confidence interval]. Each parameter showed the best prognostic performance at 0 h during the first 72 h after the ROSC. The AUC values of 0-hour CV (0.92), DV (0.84), and NPi (0.88) were comparable to that of 0-hour PLR (0.84), and the 0-hour MCV showed a lower AUC of 0.79 than 0-hour PLR. CV, constriction velocity; DV, dilation velocity; MCV, maximum constriction velocity; NPi, Neurological Pupil index.

[9] (Fig 3A, 3C and 3D). However, the 0-hour MCV showed a lower AUC of 0.79 than 0-hour PLR (Fig 3B), and MAX, MIN, and LAT yielded low prognostic value as indicated by AUC < 0.57 (S1 Fig). Combinations of 0-hour parameters with equivalent AUC values to 0-hour PLR were considered statistically unsuitable covariates in the logistic regression model because of the high correlation concerning multicollinearity.

The ROC analysis for the clinical predictors showed a high AUC value of 0.85 (Fig 4). Subsequently, we tested whether the prognostic value was improved by adding the clinical predictors to 0-hour pupillary parameters. As expected, the AUC values of the clinical predictors plus 0-hour PLR, CV, DV, or NPi reached 0.95–0.96 (Fig 4). Among 0-hour CV, DV, NPI, or PLR, only PLR remained as an independent predictor of 90-day favorable neurological

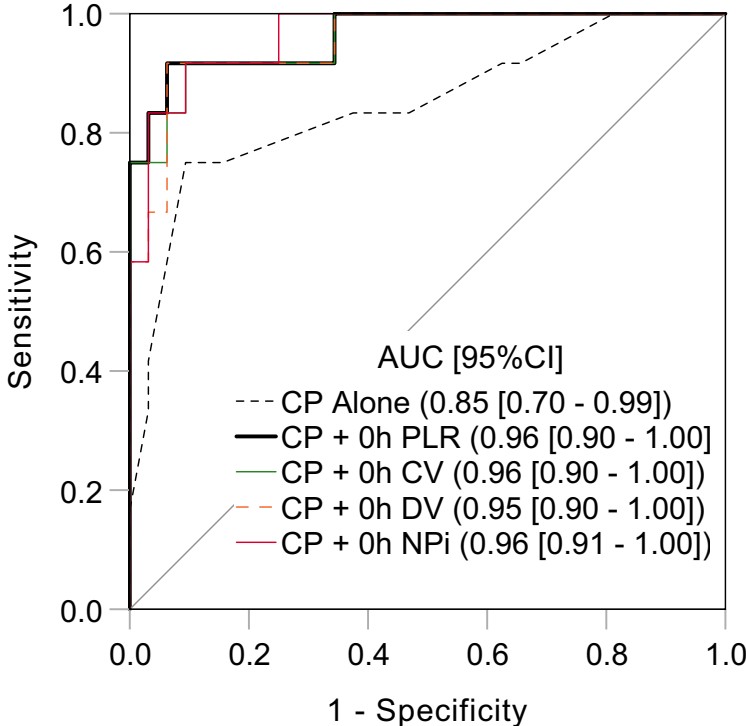

**Fig 4. Prediction of 90-day favorable neurological outcome by combination of clinical predictors and a quantitative pupillometric parameter.** Clinical parameters included in the logistic regression model were witness status, performance of bystander cardiopulmonary resuscitation, initial shockable rhythm, and implementation of target temperature management. AUC, area under the curve; CI, confidence interval; CP, clinical predictors; CV, constriction velocity; DV, dilation velocity; NPi, Neurological Pupil index; PLR, pupillary light reflex.

outcome after adjusting for known covariates (adjusted odds ratio 1.22, 95% CI [1.06–1.40], P = 0.006).

## Discussion

This study revealed that CV, MCV, DV, and NPi were consistently greater during the first 72 h after ROSC in patients who achieved favorable neurological outcome similar to PLR in our previous report [9], and they were positively correlated with PLR. Moreover, CV, DV, and NPi measured at 0 h showed an equivalent prognostic value with 0-hour PLR, and improvement of prognostic performances as high as AUC of 0.96 was achieved by combining a quantitative pupillary parameter and clinical predictors. To the best of our knowledge, this is the first comprehensive study to reveal the characteristics of quantitative pupillary responses in post-CA patients in terms of early neurological prognostication.

The clinical usefulness of the quantitative measurement of pupillary response has been reported in post-CA [5–9, 12, 13]. Most studies report an improvement of prognostic performance using quantitative PLR or NPi compared with qualitative assessment of PLR. Even though several parameters can be simultaneously obtained by a pupillometer, the characteristics of parameters other than PLR or relationship between parameters were never studied in post-CA patients. A recent study reported the relationship between CV and NPi in neurocritical care unit but did not include post-CA patients [21]. Interestingly, MAX, MIN, or LAT showed low correlation with PLR, whereas CV, MCV, or DV was strongly correlated, but NPi

was moderately correlated with PLR (Fig 2). Moreover, MAX, MIN, or LAT showing low correlation with PLR yielded low prognostic values (S1 Fig). The latency of constriction was reported to prolong with age [22] or in patients with increased intracranial pressure [16, 23]. Even though the unfavorable neurological outcome group was older in this study (Table 1), there was no difference in LAT among outcome groups (Fig 1F).

Recently, NPi was reported to show a high specificity in predicting poor neurological outcome in a large multicenter study [12]. Although the algorithm of NPi calculation is not disclosed, it is composed of several pupillary parameters showing high correlation with PLR. Thus, it is not surprising that NPi and PLR had a moderate correlation. NPi in the favorable neurological outcome group was distinctly greater with very small standard deviation than that in the unfavorable group with considerably large standard deviation (Fig 1G). This suggests that NPi may be useful for discriminating possible candidates of favorable neurological outcome rather than predicting poor neurological outcome.

Early, accurate, and simple prognostication has great clinical importance in post-CA for detecting both candidates for intensive neuroprotective therapies and irreversible anoxic brain injury for possible WLST consideration. Among the prognostic test modalities listed in the current post-CA care guidelines [4], brain computed tomography (CT) is the earliest test modality in post-CA patients with TTM. A reduced gray-to-white matter ratio (GWR) at the level of the basal ganglia on brain CT performed within 2 h after ROSC predicted a poor outcome [24, 25]. However, there are considerable variations in measurement techniques and thresholds for GWR. All prognostic physical examinations including traditional qualitative assessment of PLR are recommended to be performed at least 72 h or longer after the ROSC [4]. The guidelines also state that it is reasonable to consider persistent absence of reactivity to external stimuli and persistent burst suppression on electroencephalogram (EEG) at 72 h after ROSC and bilateral absence of the N20 somatosensory-evoked potential (SSEP) wave 24–72 h after ROSC as predictors of poor outcome [4]. However, both EEG and SSEP are not modalities that can be easily used as they require technicians to perform the test. Although neuron-specific enolase and S-100B are frequently measured, their utility is limited because of their poorly studied kinetics under TTM and unclear threshold values for early prognostication [4].

As mentioned above, since there is no established easily reproducible method to estimate the severity of brain injury and accurately predict outcomes of post-CA patients immediately after ROSC, multidisciplinary intensive care is initiated except for moribund patients. As recently stated, novel neurological prognostic measurements that can be accurate throughout a patient's recovery is necessary [26]. Therefore, the accurate prognostication of post-CA patients even immediately after ROSC has a significant clinical effect considering the time sensitivity of post-CA care. The consistency of the prognostic value of quantitative pupillary response parameters as early as 0 hours after ROSC supports this clinical utility.

The strong features of a quantitative pupillometer are its accuracy and reproducibility [27]. As recently found, values of quantitative PLR measured as continuous variables have advantages over dichotomized result of qualitative examination, which leads to information loss and unfavorable statistical properties and poorly reflects the underlying biological processes [28, 29]. Several quantitative pupillary response parameters were positively correlated with PLR, and as expected, no single quantitative pupillary response parameter was detected prominently in neurological prognostication. Thus, quantitative PLR may be the most suitable parameter because of the following reasons: it is universally measurable with different manufacturers of pupillometers, it is clinically familiar, and it has been historically studied. Other less studied quantitative pupillary response parameters that showed comparable prognostic values with 0-hour PLR warrants investigation in the future under specific pathologies. For example,

decrease in NPi may be useful for early recognition of elevated intracranial pressure in post-CA as well as in traumatic brain injury [16, 30].

We found that a single 0-hour pupillary response parameter shows better prognostic value of 90-day favorable neurological outcome than a combination of patient demographics and treatment information. Further, by combining this clinical information and a 0-hour pupillary response parameter, a prognostic performance with the highest AUC value of 0.96 could be achieved. In an exploratory analysis using a multivariable regression model, 0-hour PLR remained as an independent predictor of 90-day favorable neurological outcome after adjusting for resuscitation and post-resuscitation characteristics. Further investigations are warranted to elucidate the role of 0-hour PLR in multimodality early prognostic bundle.

This study has several limitations. First, the sample size was small. Since quantitative measurement immediately after ROSC was one of the major foci of this study, a substantial number of patients were excluded because of the difficulty of obtaining informed consent from an accompanying family member immediately after ROSC. However, our study has a multicenter design, which avoids biases inherent to previous single-center studies. Second, since this was an observational study and post-arrest care was performed according to a standardized institutional treatment protocol, adjunctive prognostic tests were not standardized. Thus, we did not test the prognostic value of the quantitative pupillary parameters as in a multimodal prognostication bundle with other physiological or biomarker tests, and we could not include all potential confounders for adjustment in the logistic regression model. Third, treating physicians were not blinded to the results of quantitative pupillary responses. Japanese physicians seldom adopt active WLST even if poor neurological prognosis is perceived. As mortality continued to increase over 3 weeks and did not concentrate in the first week [9], we consider that unblinded results have had small effect on self-fulfilling prophecy.

## Conclusions

Several additional parameters were associated with favorable neurologic outcomes including CV, MCV, DV, and NPi, similar to PLR. These parameters were consistently greater in the favorable neurological outcome group of post-CA patients during the first 72 h after ROSC and showed positive correlation with PLR. CV, DV, or NPi showed equivalent prognostic performance to PLR at 0 h after ROSC independently. We hope that future studies will reveal the utility of 0-hour quantitative pupillary response, especially PLR as a parameter for multimodal prognostication.

## Supporting information

**S1 Table. Median values of quantitative pupillary parameters.** CV, constriction velocity; DV, dilation velocity; IQR, interquartile range; LAT, latency of constriction; MAX, maximum diameter; MCV, maximum constriction velocity; MIN, minimum diameter; NPi, neurological pupil index.
(DOCX)

**S2 Table. Minimal dataset.**
(XLSX)

**S1 Fig. Serially measured quantitative pupillary responses of favorable and unfavorable neurological outcomes after return of spontaneous circulation to 72 h.** Values represent area under the curve [95% confidence interval]. LAT, latency of constriction; MAX, maximum diameter; MIN, minimum diameter.
(DOCX)

## Acknowledgments

The authors thank Sayuri Suzuki, PhD, for assistance with data collection.

## Author Contributions

**Conceptualization:** Tomoyoshi Tamura, Jun Namiki, Takayuki Abe.

**Data curation:** Tomoyoshi Tamura, Yoko Sugawara, Kazuhiko Sekine, Kikuo Yo, Takahiro Kanaya, Shoji Yokobori.

**Formal analysis:** Takayuki Abe.

**Funding acquisition:** Jun Namiki.

**Investigation:** Tomoyoshi Tamura, Jun Namiki, Yoko Sugawara.

**Methodology:** Tomoyoshi Tamura, Jun Namiki, Hiroyuki Yokota.

**Project administration:** Tomoyoshi Tamura, Jun Namiki, Junichi Sasaki.

**Supervision:** Jun Namiki, Kazuhiko Sekine, Kikuo Yo, Shoji Yokobori, Hiroyuki Yokota, Junichi Sasaki.

**Writing – original draft:** Tomoyoshi Tamura.

**Writing – review & editing:** Jun Namiki, Yoko Sugawara, Kazuhiko Sekine, Kikuo Yo, Takahiro Kanaya, Shoji Yokobori, Takayuki Abe, Hiroyuki Yokota, Junichi Sasaki.

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
