## [Decision Letter · Decision Letter 0]

4 Nov 2019

PONE-D-19-28265

Early outcome prediction with quantitative pupillary response parameters after out-of-hospital cardiac arrest: A multicenter prospective observational study

PLOS ONE

Dear Dr Namiki,

Thank you for submitting your manuscript to PLOS ONE. After careful consideration, we feel that it has merit but does not fully meet PLOS ONE’s publication criteria as it currently stands. Therefore, we invite you to submit a revised version of the manuscript that addresses the points raised during the review process.

We would appreciate receiving your revised manuscript by December 1, 2019. To enhance the reproducibility of your results, we recommend that if applicable you deposit your laboratory protocols in protocols.io, where a protocol can be assigned its own identifier (DOI) such that it can be cited independently in the future. For instructions see: http://journals.plos.org/plosone/s/submission-guidelines#loc-laboratory-protocols

We look forward to receiving your revised manuscript.

Kind regards,

Steve Lin

Academic Editor

PLOS ONE

Journal Requirements:

"This study is in adherence with ethical guidelines and approved by IRB of each participating institution. Written informed consent was obtained from patients’ family as appropriate. ".

Additional Editor Comments (if provided):

Thank you for submitting this manuscript. It is an interesting study. There were several comments and suggestions that our reviewers have raised. Please address these to improve the manuscript for publication.

Reviewers' comments:

Reviewer's Responses to Questions

**Comments to the Author**

1. Is the manuscript technically sound, and do the data support the conclusions?

Reviewer #1: Yes

Reviewer #2: Yes

2. Has the statistical analysis been performed appropriately and rigorously? 

Reviewer #1: Yes

Reviewer #2: Yes

3. Have the authors made all data underlying the findings in their manuscript fully available?

Reviewer #1: Yes

Reviewer #2: No

4. Is the manuscript presented in an intelligible fashion and written in standard English?

Reviewer #1: Yes

Reviewer #2: Yes

5. Review Comments to the Author

Reviewer #1: Thank you for the opportunity to review your manuscript. The study design is adequate and tests an interesting hypothesis.

Could the study authors comment on why the 0 hour measurements were the only measurements found to significantly impact the ROCs?

I find the heterogeneity of the patient characteristics and small sample size problematic in being able to exclude patient characteristics as a main driver in the quantitative pupillary response differences at time 0. The authors comment on pupillary response differences based on age, drugs used during resuscitation and therapeutic hypothermia. Could the authors comment on why the differences noted in pupillary responses between the two groups are not accounted for by the differences in patient characteristics?

Could the authors also clarify the statement on page 8 line 168-171: "Similar to PLR (P=0.86)[9], after accounting for the interaction at a certain time point by outcome status, none of the pupillary parameters were significant...". If this statement refers to the fact that there was no difference in change of pupillary parameters over time regardless of outcome, then the statement would need to be reworded to reflect this.

Reviewer #2: General comments: While outcome prediction from cardiac arrest using pupillometry is no longer a novel concept, this study adds important data regarding 1) optimal timing of the tool and 2) thorough assessment of alternate predictors. These data will be useful as the tool becomes more prevalent at the bedside. In this way, the study will be of interest to PLOS ONE readers. However, a key limitation of the study that the authors should address is the lack of transparent and detailed reporting of patient CPC, which will affect how their results should be interpreted.

Methods: Why did you choose to include non-comatose post-CA patients? To reference your introduction, the literature for PLR is in comatose post-CA patients. The description of the protocol (line 94 to end of paragraph) also leads the reader to think this study was conducted in comatose patients; please clarify the language.

Analysis: For the selection of potential confounders, did you include CPR duration? It is not listed in line 136. Was the “a priori knowledge” of selected clinical predictors based on clinical expertise or a prior logistic regression model? Please expand.

Results: Please report the breakdown of CPC. How many patients were in each CPC category, (i.e., 1, 2, 3, 4, 5)? This is relevant because your study’s two outcome groups could be weighted towards one extreme. For example, the “unfavorable neurologic outcome” group could be mostly CPC 5 (brain death) or perhaps CPC 3 (severe but conscious). These are clearly two very different phenotypes. I would interpret the study differently depending on the breakdown of the groups. Also, it is standard to report CPC subgroups. Additionally, for quantitative measurements of pupillary responses (line 163), please report median values and IQR in the text, or include these values as a supplemental table.

Discussion: Were there any outliers? It may be useful to discuss if there were patients who had a low pupillary response but who still had a favorable neurologic outcome. Regarding line 319, I think you mean to say “Japanese physicians DO NOT withhold aggressive treatments.” It is important to mention that the study was unblinded, and I think while the cultural insights here are very useful to the reader, the language in the last sentence line 319 reads a bit too confident.

Conclusions: This is alphabet soup and a bit too granular. You might start the paragraph by saying “several additional parameters were associated with favorable neurologic outcome, including XYZ.” The final sentence line 325 is confusing and not grammatically correct and needs to be reworked.

Figures: Figure 1: please show subgroups (n=x) of CPC. It appears there is overlap between groups (e.g., frames C, D, E, F, G) but the SEM are not reported in the text. Please specify in the results section. General comments: would be helpful for the reader if you label each panel more clearly.

Tables: Please report CPC subgroups.

6. PLOS authors have the option to publish the peer review history of their article (what does this mean?). If published, this will include your full peer review and any attached files.

Reviewer #1: Yes: Alina Toma

Reviewer #2: No

---

## [Author Response · Author response to Decision Letter 0]

30 Nov 2019

Please find our point-by-point response to each comment in the separate file.

---

## [Decision Letter · Decision Letter 1]

10 Jan 2020

Early outcome prediction with quantitative pupillary response parameters after out-of-hospital cardiac arrest: A multicenter prospective observational study

PONE-D-19-28265R1

Dear Dr. Namiki,

We are pleased to inform you that your manuscript has been judged scientifically suitable for publication and will be formally accepted for publication once it complies with all outstanding technical requirements.

With kind regards,

Steve Lin

Academic Editor

PLOS ONE

Additional Editor Comments (optional):

Reviewers' comments:

Reviewer's Responses to Questions

**Comments to the Author**

1. If the authors have adequately addressed your comments raised in a previous round of review and you feel that this manuscript is now acceptable for publication, you may indicate that here to bypass the “Comments to the Author” section, enter your conflict of interest statement in the “Confidential to Editor” section, and submit your "Accept" recommendation.

Reviewer #1: All comments have been addressed

Reviewer #2: All comments have been addressed

2. Is the manuscript technically sound, and do the data support the conclusions?

Reviewer #1: Yes

Reviewer #2: Yes

3. Has the statistical analysis been performed appropriately and rigorously? 

Reviewer #1: Yes

Reviewer #2: Yes

4. Have the authors made all data underlying the findings in their manuscript fully available?

Reviewer #1: Yes

Reviewer #2: Yes

5. Is the manuscript presented in an intelligible fashion and written in standard English?

Reviewer #1: Yes

Reviewer #2: Yes

6. Review Comments to the Author

Reviewer #1: Thank you for addressing all the concerns. I feel this paper is ready for publication in its current form.

Reviewer #2: (No Response)

7. PLOS authors have the option to publish the peer review history of their article (what does this mean?). If published, this will include your full peer review and any attached files.

Reviewer #1: No

Reviewer #2: No

---

## [Editor Report · Acceptance letter]

11 Feb 2020

PONE-D-19-28265R1 

Early outcome prediction with quantitative pupillary response parameters after out-of-hospital cardiac arrest: A multicenter prospective observational study 

Dear Dr. Namiki:

I am pleased to inform you that your manuscript has been deemed suitable for publication in PLOS ONE. Congratulations! Your manuscript is now with our production department. 

With kind regards,

on behalf of

Dr. Steve Lin 

Academic Editor

PLOS ONE